# The tardigrade damage suppressor protein binds to nucleosomes and protects DNA from hydroxyl radicals

**Carolina Chavez[†], Grisel Cruz-Becerra[†], Jia Fei[†], George A Kassavetis[†], James T Kadonaga***

Section of Molecular Biology, University of California, San Diego, San Diego, United States

**Abstract** Tardigrades, also known as water bears, are animals that can survive extreme conditions. The tardigrade *Ramazzottius varieornatus* contains a unique nuclear protein termed Dsup, for damage suppressor, which can increase the resistance of human cells to DNA damage under conditions, such as ionizing radiation or hydrogen peroxide treatment, that generate hydroxyl radicals. Here we find that *R. varieornatus* Dsup is a nucleosome-binding protein that protects chromatin from hydroxyl radicals. Moreover, a Dsup ortholog from the tardigrade *Hypsibius exemplaris* similarly binds to nucleosomes and protects DNA from hydroxyl radicals. Strikingly, a conserved region in Dsup proteins exhibits sequence similarity to the nucleosome-binding domain of vertebrate HMGN proteins and is functionally important for nucleosome binding and hydroxyl radical protection. These findings suggest that Dsup promotes the survival of tardigrades under diverse conditions by a direct mechanism that involves binding to nucleosomes and protecting chromosomal DNA from hydroxyl radicals.

DOI: https://doi.org/10.7554/eLife.47682.001

**\*For correspondence:**
jkadonaga@ucsd.edu

[†]These authors contributed equally to this work

**Competing interests:** The authors declare that no competing interests exist.

## Introduction

Tardigrades, which are also known as water bears or moss piglets, are small invertebrate animals that are found in marine, freshwater, and terrestrial habitats throughout the Earth (reviewed in *Guidetti et al., 2012*; *Møbjerg et al., 2011*; *Weronika and Łukasz, 2017*). They are typically about 0.1 to 1 mm in length, and comprise a head segment in addition to four body segments that each contains two legs with claws. Terrestrial tardigrades require a thin film of water to remain active. In the absence of water, they undergo anhydrobiosis into a dormant dehydrated state from which they can be rehydrated to an active form. In the anhydrobiotic state, tardigrades are resistant to extreme conditions of heat, cold, vacuum, pressure, radiation, and chemical treatments. Remarkably, they have been found to survive exposure to the vacuum and radiation of outer space (*Jönsson et al., 2008*). Thus, the unique properties of tardigrades have led to considerable interest in these animals. The analysis of their singular features should lead to significant new biological insights.

The molecular analysis of tardigrades has been advanced by the sequencing of the genomes of *Ramazzottius varieornatus* (*Hashimoto et al., 2016*) and *Hypsibius exemplaris* (*Yoshida et al., 2017*). [Note: the strain of tardigrades that was designated as *Hypsibius dujardini* in *Yoshida et al. (2017)* has since been found to be a new species that is now termed *Hypsibius exemplaris* (*GĄsiorek et al., 2018*). We will use the new terminology in this paper.] In *R. varieornatus*, the study of chromatin-associated factors revealed a tardigrade-specific protein, termed Dsup, for damage suppressor (*Hashimoto et al., 2016*; *Hashimoto and Kunieda, 2017*). Dsup is a highly charged and largely unstructured nuclear protein that binds to DNA. Intriguingly, the expression of Dsup gene in human cells was observed to decrease DNA fragmentation induced by X-ray irradiation or treatment

**eLife digest :** Tardigrades, also known as water bears and moss piglets, are small animals found in many different environments on land and sea. These animals have the remarkable ability to survive extremes including very low temperatures, high levels of radiation and exposure to chemicals that are harmful to other forms of life. Tardigrades have even been found to survive the harsh conditions of outer space.

X-rays are a type of radiation naturally produced by lightning strikes and are also found in cosmic rays from outer space. High doses of X-rays can cause genetic mutations that may lead to serious illness or death. This is because when X-rays come into contact with water they split the water molecules to make particles known as hydroxyl radicals, which in turn damage the DNA inside cells.

The genomes of animals and plants are made of DNA, which is packaged into a structure called chromatin. Previous studies identified a protein named Dsup in a tardigrade called *Ramazzottius varieornatus* that can protect human cells from damage by X-rays. However, it was not known whether Dsup binds directly to chromatin or plays a more indirect role in protecting DNA.

Chavez, Cruz-Becerra, Fei, Kassavetis et al. used biochemical approaches to study Dsup. Their experiments revealed that Dsup from *R. varieornatus* binds to chromatin to protect the DNA from damage by hydroxyl radicals, and that the Dsup protein in another tardigrade species also works in a similar way. Further analysis showed that a region of Dsup that is needed to bind to chromatin is very similar to a region that had been previously found only in chromatin-binding proteins from humans and other vertebrates (animals with backbones). This connection between Dsup and vertebrate chromatin-binding proteins remains a mystery.

The new findings about tardigrade Dsup may help researchers develop animal cells that live longer under normal or extreme environmental conditions. In this manner, Dsup could be used to expand the range of applications of cells in biotechnology. It could also increase the effectiveness of current methods, such as the production of some pharmaceuticals, that depend upon the use of cultured cells.

DOI: https://doi.org/10.7554/eLife.47682.002

with hydrogen peroxide. Moreover, Dsup-containing human cells exhibited higher viability after X-ray irradiation than control cells. These findings suggest that Dsup is a unique protein that either directly or indirectly protects DNA.

In this work, we sought to examine the molecular function of Dsup. It was previously shown that Dsup interacts with free DNA in an apparently nonspecific manner (*Hashimoto et al., 2016*). Because the natural form of DNA in the nucleus is chromatin, we investigated the binding of Dsup to nucleosomes. These experiments revealed that *R. varieornatu*s Dsup binds preferentially to nucleosomes relative to free DNA. Then, to test the relevance of these findings to other tardigrades, we examined the properties of a Dsup-like protein from *H. exemplaris*, and found that this protein appears to be an ortholog of *R. varieornatus* Dsup. Intriguingly, the Dsup proteins contain a region with sequence similarity to core consensus of the nucleosome-binding domain of vertebrate high mobility group N (HMGN) proteins, and the HMGN-like sequence in Dsup is important for its binding to nucleosomes. Furthermore, in a purified biochemical system, both Dsup proteins are able to protect chromatin from cleavage by hydroxyl radicals, which are generated in cells by ionizing radiation as well as by treatment with hydrogen peroxide. These studies thus reveal conserved functions by which Dsup maintains the integrity of chromosomal DNA under extreme conditions.

## Results

### *R. varieornatus* Dsup binds preferentially to mononucleosomes relative to free DNA

The *R. varieornatus* Dsup protein is a largely disordered and highly charged nuclear protein (*Figure 1A*) (*Hashimoto et al., 2016*; *Hashimoto and Kunieda, 2017*). To study its biochemical properties, we synthesized a FLAG- and His6-tagged version of the full-length protein in *Escherichia*

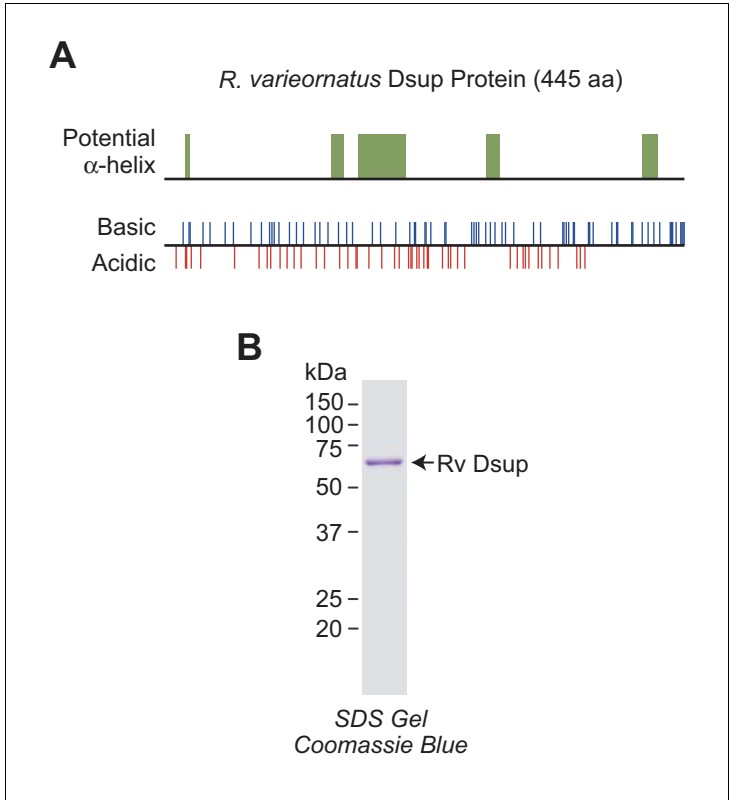

**Figure 1.** *R.varieornatus* damage suppressor protein (Rv Dsup). (**A**) Rv Dsup is a highly charged protein. The protein was analyzed by using the MPI Bioinformatics Toolkit (*Zimmermann et al., 2018*), and it is predicted to be disordered except for some potentially alpha-helical regions (upper panel). The positions of basic and acidic amino acid residues are also shown (lower panel). (**B**) Purification of recombinant Rv Dsup protein synthesized in *Escherichia coli*. The recombinant Rv Dsup has an N-terminal His6-tag and a C-terminal FLAG tag. The purified protein was analyzed by 12% polyacrylamide-SDS gel electrophoresis and staining with Coomassie Brilliant Blue R-250. The sizes of molecular mass markers (in kDa) are indicated.
DOI: https://doi.org/10.7554/eLife.47682.003

coli and purified it to greater than 95% homogeneity by affinity chromatography (*Figure 1B*). We will refer to the *R. varieornatus* Dsup protein as 'Rv Dsup'.

Although Rv Dsup was observed to associate with free DNA (*Hashimoto et al., 2016*), we tested the binding of Rv Dsup to nucleosomes, which are the natural form of DNA in the eukaryotic nucleus. To this end, we reconstituted mononucleosomes with the 5S rDNA nucleosome positioning sequence from *Xenopus borealis* (*Rhodes, 1985*; *Hayes et al., 1990*; *Fei et al., 2018*). We compared the binding of Rv Dsup to 147 bp DNA-containing mononucleosomes relative to the corresponding 147 bp free DNA by gel mobility shift analysis (*Figure 2A* and *Figure 2—figure supplement 1A*). These experiments revealed that Rv Dsup binds with a higher affinity to nucleosomes than to free DNA.

To test whether this effect is specific for the 5S rDNA sequence, we examined the binding of Rv Dsup to mononucleosomes and free DNA that contain the 601 nucleosome positioning sequence (*Lowary and Widom, 1998*). As with the 5S rDNA nucleosomes, we observed more efficient binding of Rv Dsup to the 601 mononucleosomes than to the 601 free DNA (*Figure 2B* and *Figure 2—figure supplement 1B*). Thus, Rv Dsup binds preferentially to mononucleosomes relative to free DNA in a manner that appears to be independent of the specific DNA sequence.

Because the 147 bp DNA-containing mononucleosomes lack linker DNA that extends beyond the nucleosome core, we compared the binding of Rv Dsup to mononucleosomes that contain either 147 bp DNA or 181 bp DNA (*Figure 2C*). These experiments showed that the presence of linker

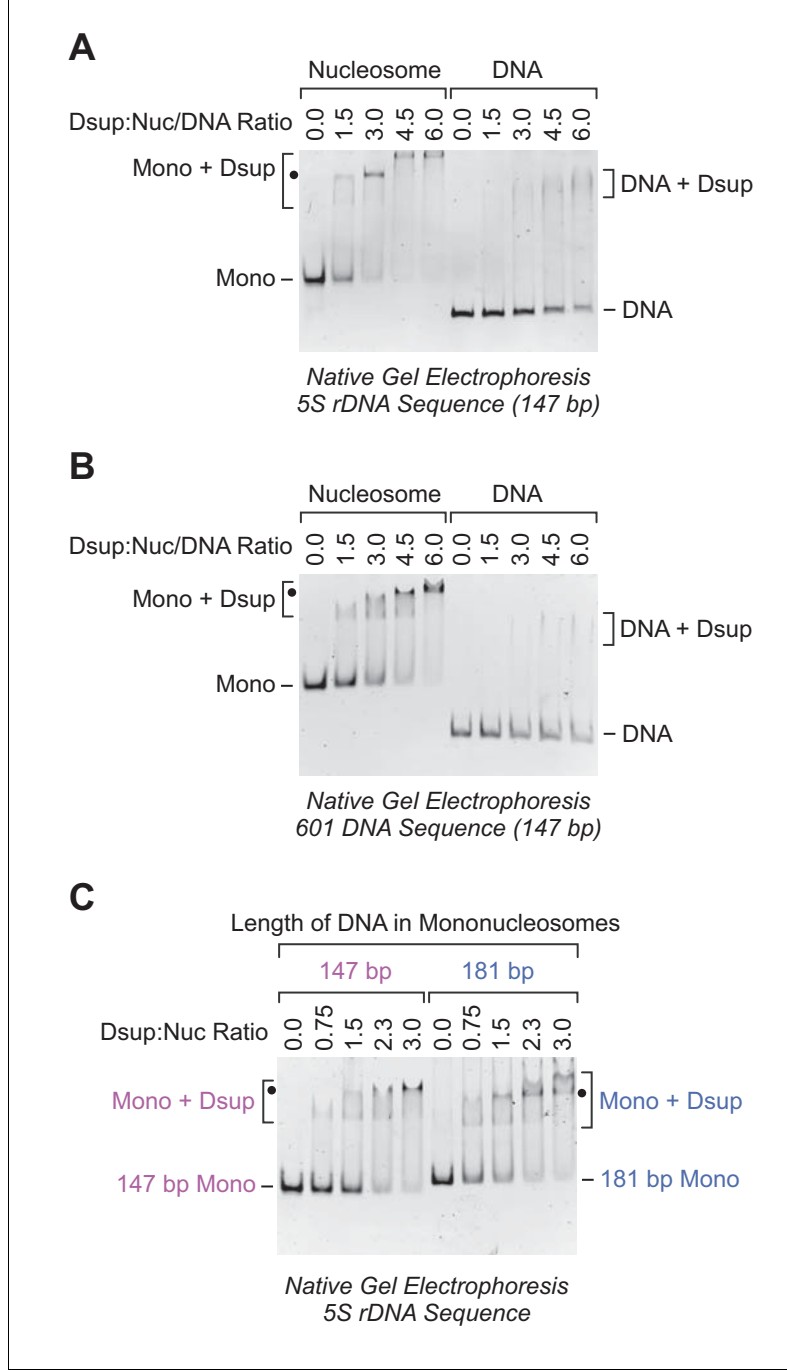

**Figure 2.** Rv Dsup binds preferentially to mononucleosomes relative to free DNA. (**A**) Gel mobility shift analysis with the *Xenopus borealis* 5S rDNA (147 bp) as either mononucleosomes or free DNA. The indicated amounts of purified Rv Dsup was combined with the mononucleosomes or DNA, and the resulting samples were subjected to nondenaturing 4.5% polyacrylamide gel electrophoresis and staining with ethidium bromide. The positions of mononucleosomes (*Mono*), Rv Dsup-nucleosome complexes (*Mono + Dsup*; the major shifted distinct band is denoted by a *black dot*), free DNA fragments (*DNA*), and Rv Dsup-DNA complexes (*DNA + Dsup*) are shown. (**B**) Gel mobility shift analysis of Rv Dsup with the 601 synthetic DNA sequence (147 bp) as either mononucleosomes or free DNA. Reactions were performed as in A. (**C**) Effect of DNA length upon binding of Rv Dsup to mononucleosomes. Gel mobility shift assays were performed with mononucleosomes, as in A, with 5S rDNA sequences with a length of either 147 bp or 181 bp.

DOI: https://doi.org/10.7554/eLife.47682.004

*Figure 2 continued on next page*

*Figure 2 continued*

The following figure supplement is available for figure 2:

**Figure supplement 1.** Quantitation of gel mobility shift analyses of Rv Dsup binding to mononucleosomes or free DNA.

DOI: https://doi.org/10.7554/eLife.47682.005

DNA did not substantially alter the efficiency of Rv Dsup binding to nucleosomes. It thus appears that Rv Dsup binds primarily to the nucleosome core rather than to the linker DNA.

In the gel shift experiments with Rv Dsup and mononucleosomes, there is typically a distinct major shifted band (denoted by black dots in *Figure 2*) along with a minor faster migrating band. This pattern can be seen particularly clearly in the titration of Rv Dsup with 147 bp 5S rDNA mononucleosomes in *Figure 2C*, and is suggestive of the binding of two molecules of Rv Dsup to the nucleosome. With the 5S rDNA mononucleosomes and high concentrations of Rv Dsup, we also observed a slower migrating band (*Figure 2A*), which may reflect additional Rv Dsup binding to the nucleosomes and/or higher-order aggregation of the Rv Dsup-nucleosome complexes. In contrast, a distinct shifted band is not seen with Rv Dsup and free DNA.

## Dsup is incorporated into periodic nucleosome arrays

To complement the studies with mononucleosomes, we tested whether Rv Dsup can be incorporated into extended periodic nucleosome arrays. In these experiments, we assembled nucleosomes onto plasmid DNA by using a purified and defined ATP-dependent chromatin assembly system with the ACF motor protein and the dNLP core histone chaperone (*Fyodorov and Kadonaga, 2003*; *Khuong et al., 2017*). We also included separate reactions with histone H1 (reviewed in *Woodcock et al., 2006*; *Happel and Doenecke, 2009*; *Kalashnikova et al., 2016*) as a positive control for a nucleosome-binding factor.

The presence of Rv Dsup does not inhibit the efficiency of ACF-mediated chromatin assembly, as measured by the DNA supercoiling assay (*Figure 3A*). Partial MNase digestion analysis showed that the assembly of chromatin with Rv Dsup results in a small but distinct increase in the nucleosome repeat length (*Figure 3B*). To analyze the binding of Rv Dsup to the nucleosome arrays, we extensively digested the ACF-assembled chromatin with MNase, and subjected the resulting mononucleosome species to nondenaturing gel electrophoresis (*Figure 3C*). This experiment revealed that Rv Dsup-bound mononucleosome particles are released from the chromatin upon extensive MNase digestion. We then compared the Rv Dsup-mononucleosome particles that were generated via ACF assembly followed by MNase digestion, as in *Figure 3C*, with the Rv Dsup-mononucleosome particles formed by the addition of Rv Dsup to mononucleosomes, as in *Figure 2A*, and found that the differently prepared Rv Dsup-nucleosome particles migrate at approximately the same rate during nondenaturing gel electrophoresis (*Figure 3D*). It therefore appears that the interaction of Rv Dsup with nucleosomes in periodic arrays, as in *Figure 3*, is similar to the binding of Rv Dsup to mononucleosome particles, as in *Figure 2*.

## Dsup and histone H1 can bind simultaneously to nucleosomes

Histone H1 is a major nucleosome-binding protein in animals, and it has been found to be present at approximately one molecule per nucleosome (*Bates and Thomas, 1981*) (reviewed in *Woodcock et al., 2006*; *Happel and Doenecke, 2009*; *Kalashnikova et al., 2016*). We were therefore interested in testing whether histone H1 and Rv Dsup can bind simultaneously to nucleosomes. To this end, we carried out gel mobility shift analyses with 181 bp DNA mononucleosomes in the presence of varying concentrations of Rv Dsup and histone H1 (*Figure 4A*). (We used 181 bp DNA mononucleosomes to allow the binding of histone H1 to the linker DNA.) These experiments revealed that the addition of Rv Dsup to H1-bound mononucleosomes results in the formation of a slower migrating Rv Dsup-H1-nucleosome complex. Likewise, the addition of histone H1 to Rv Dsup-bound mononucleosomes yields Rv Dsup-H1-nucleosome complexes that migrate at the same rate as the species formed upon addition of Rv Dsup to H1-bound nucleosomes. These results indicate that histone H1 and Rv Dsup can bind simultaneously to mononucleosomes.

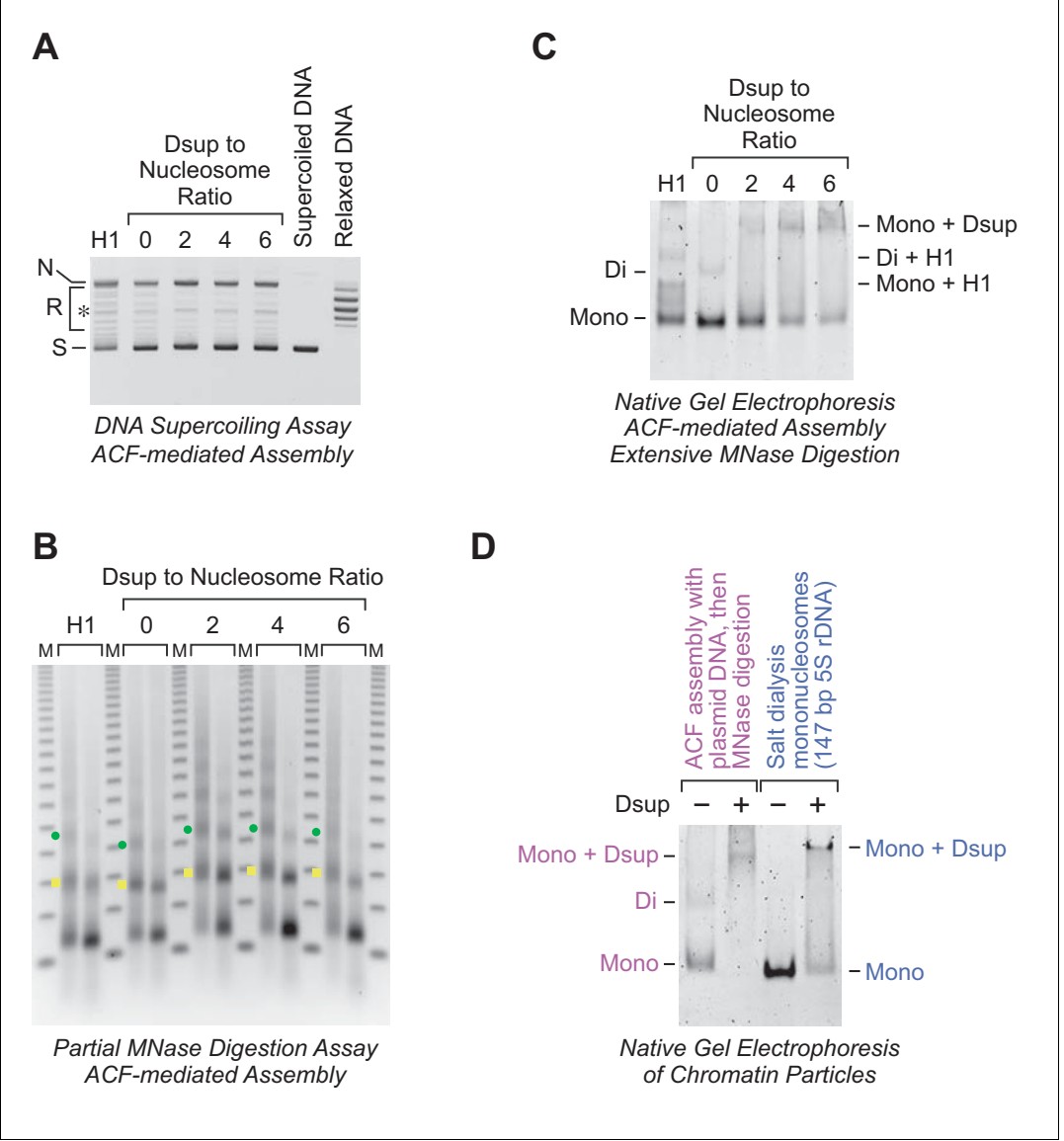

**Figure 3.** Rv Dsup can be incorporated into periodic nucleosome arrays. The ATP-dependent assembly of periodic nucleosome arrays was carried out with purified ACF, dNLP, core histones, ATP, relaxed plasmid DNA, and topoisomerase I (*Fyodorov and Kadonaga, 2003*; *Khuong et al., 2017*) in the absence or presence of purified Rv Dsup. (**A**) DNA supercoiling analysis indicates that Rv Dsup does not affect the efficiency of nucleosome assembly. Chromatin assembly reactions were performed with the indicated amounts of purified Rv Dsup. The reaction products were deproteinized and subjected to 0.8% agarose gel electrophoresis, and the DNA species were visualized by staining with ethidium bromide. Samples of supercoiled DNA and relaxed DNA were included as references. The positions of nicked DNA (N), relaxed DNA (R), and supercoiled DNA (S) are indicated. The asterisk denotes a minor amount of linear DNA that is generated by a nuclease contaminant in the chromatin assembly factors. (**B**) Partial MNase digestion analysis reveals an increase in the nucleosome repeat length upon incorporation of Rv Dsup into chromatin. Chromatin assembly reactions were performed as in A, and the resulting samples were partially digested with two different concentrations of MNase, deproteinized, and subjected to 1.3% agarose gel electrophoresis. Reactions with purified *Drosophila* histone H1 were included as a reference. The *yellow squares* denote the DNA fragments derived from dinucleosomes, and the *green dots* correspond to the DNA fragments derived from trinucleosomes. The DNA size markers (M) are the 123 bp ladder (Invitrogen). (**C**) Native gel electrophoresis of mono- and dinucleosome particles obtained from extensive MNase digestion of chromatin that is assembled in the absence or presence of Rv Dsup. Reactions were performed as in A and B except that the products were subjected to extensive MNase digestion followed by nondenaturing 4.5% polyacrylamide gel electrophoresis. Reactions with histone H1 were included as a reference. The chromatin

*Figure 3 continued on next page*

*Figure 3 continued*
particles were detected by staining with ethidium bromide. The positions of mononucleosomes (*Mono*), dinucleosomes (*Di*), monucleosomes with Rv Dsup (*Mono + Dsup*), mononucleosomes with H1 (*Mono + H1*), and dinucleosomes with H1 (*Di + H1*) are indicated. (**D**) Dsup-mononucleosome particles generated by ACF assembly followed by MNase digestion migrate on native gels at approximately the same rate as Dsup-mononucleosome particles formed by the addition of Dsup to mononucleosomes. Native gel electrophoresis of chromatin particles was performed with salt dialysis-reconstituted mononucleosomes containing Rv Dsup (as in *Figure 2A*) as well as with mononucleosomes generated by MNase digestion of Rv Dsup-containing chromatin assembled with ACF (as in C). The positions of the mononucleosomes (*Mono*) and mononucleosome-Rv Dsup complexes (*Mono +Dsup*) are denoted.

DOI: https://doi.org/10.7554/eLife.47682.006

We then examined whether the presence of both histone H1 and Rv Dsup results in a disruption of chromatin structure. To address this question, we carried out ACF-mediated chromatin assembly reactions in the presence or absence of histone H1 and/or Rv Dsup. These experiments revealed that the addition of both H1 and Rv Dsup did not alter the efficiency of chromatin assembly, as measured by the DNA supercoiling assay (*Figure 4B*). Furthermore, the presence of histone H1 and Rv Dsup did not disrupt the periodicity of the nucleosome arrays, as assessed with the partial MNase digestion assay (*Figure 4C*). Thus, histone H1 and Rv Dsup can bind simultaneously to nucleosomes,

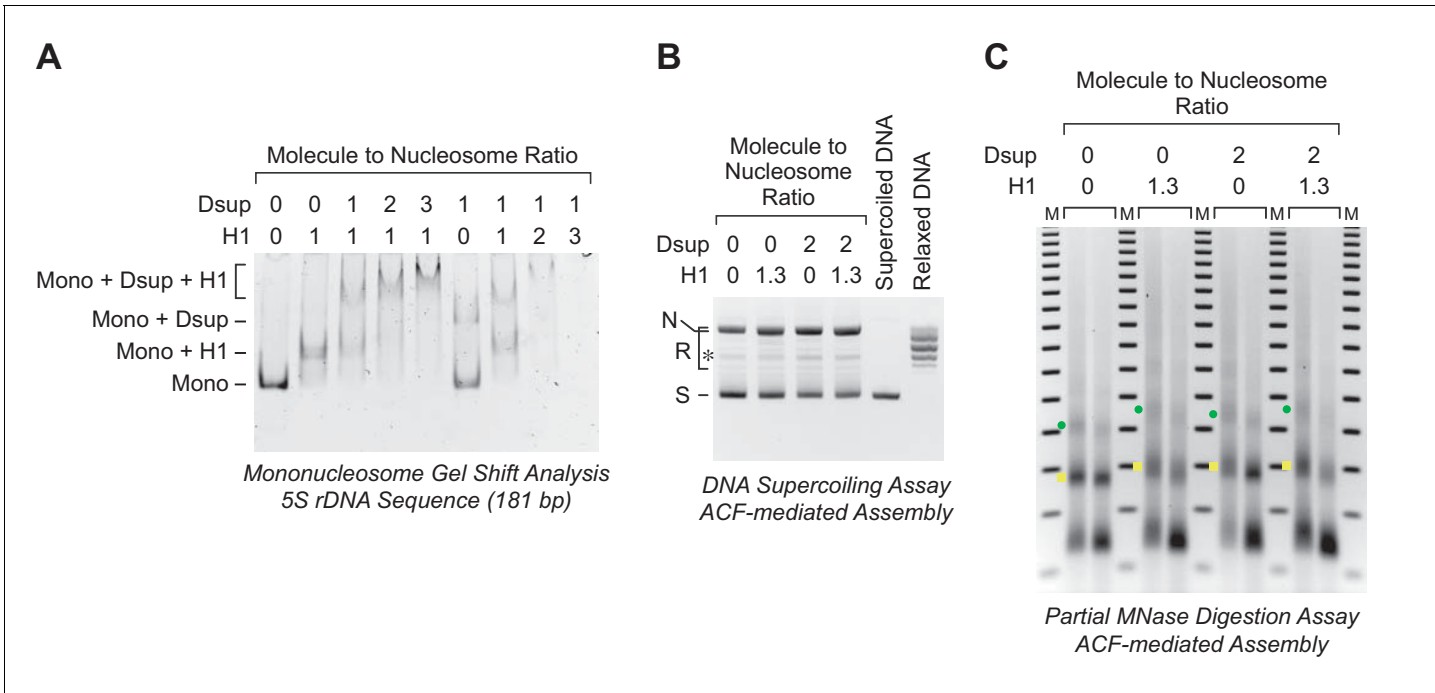

**Figure 4.** Rv Dsup and histone H1 can bind simultaneously to nucleosomes. (**A**) Native gel mobility shift analysis reveals the binding of both Rv Dsup and histone H1 to mononucleosomes. Experiments were performed with 5S rDNA mononucleosomes (181 bp DNA) and the indicated amounts of Rv Dsup and *D. melanogaster* histone H1. The samples were subjected to nondenaturing 4.5% polyacrylamide gel electrophoresis, and the DNA was visualized by staining with ethidium bromide. The positions of mononucleosomes (*Mono*), mononucleosome-H1 complexes (*Mono + H1*), mononucleosome-Rv Dsup complexes (*Mono + Dsup*), and mononucleosome-Rv Dsup-H1 complexes (*Mono + Dsup + H1*) are shown. The loss of signal at the highest concentration of H1 appears to be due to H1-mediated aggregation of the sample, as is typically seen with excess H1 (see, for example, *Hashimoto et al., 2016*). (**B**) DNA supercoiling analysis indicates that efficiency of nucleosome assembly is not significantly altered by the presence of Rv Dsup and histone H1. Chromatin assembly reactions were performed and analyzed as in *Figure 3A*. A minor amount of linear DNA is indicated by an asterisk. (**C**) Partial MNase digestion analysis shows that the nucleosome repeat length in the presence of histone H1 and Rv Dsup is similar to that seen with histone H1 alone. Chromatin assembly reactions were performed and analyzed as in *Figure 3B*. The *yellow squares* denote the DNA fragments derived from dinucleosomes, and the *green dots* correspond to the DNA fragments derived from trinucleosomes.

DOI: https://doi.org/10.7554/eLife.47682.007

and this binding does not substantially alter or disrupt chromatin structure. These findings are consistent with the observation that Rv Dsup is not deleterious to histone H1-containing human cells (*Hashimoto et al., 2016*).

## The Dsup-like protein in *H. exemplaris* is an ortholog of *R. varieornatus* Dsup

It was also important to assess whether Dsup is present in tardigrades other than *R. varieornatus*. In this regard, we were interested in *H. exemplaris*, a tardigrade that was recently subjected to genome resequencing (*Yoshida et al., 2017*). *R. varieornatus* and *H. exemplaris* are limnoterrestrial tardigrades in the family Hypsibiidae, and both animals can undergo anhydrobiosis (see, for example, *Horikawa et al., 2008*; *Kondo et al., 2015*; *Wright, 1989*). Dsup was initially not identified in *H. exemplaris* (*Yoshida et al., 2017*), but further analysis revealed a Dsup-like protein in this organism (*Hashimoto and Kunieda, 2017*). The *H. exemplaris* Dsup-like protein was found to have 26.4% amino acid identity with Rv Dsup as well as hydrophobicity and charge profiles that are similar to those of Rv Dsup.

To investigate the evolutionary relationship between these proteins, we examined the chromosomal regions that encompass the Dsup-related genes in *R. varieornatus* and *H. exemplaris*. This analysis revealed that the Dsup-related genes in both organisms are flanked by the same neighboring genes (*Figure 5A*). Hence, based on the similarity of the proteins, including the closely related nuclear localization signals near the C-termini (*Hashimoto and Kunieda, 2017*), and the similarity of the gene arrangements (*Figure 5A*), it appeared likely that the *H. exemplaris* Dsup-like protein and *R. varieornatus* Dsup are homologs.

It remained to be determined, however, whether the two proteins have related biochemical functions. To this end, we synthesized and purified the *H. exemplaris* Dsup-like protein, which we will refer to as 'He Dsup' (*Figure 5B*). We then tested the binding of He Dsup to mononucleosomes and to free DNA, and found that He Dsup binds preferentially to nucleosomes relative to free DNA (*Figure 5C*), as does Rv Dsup (*Figure 2*). In addition, we directly compared the binding of He Dsup and Rv Dsup to mononucleosomes, and observed that each of the proteins shifts mononucleosomes to approximately the same position on the gel (*Figure 5D*). The slower migration of the Rv Dsup-nucleosome complexes relative to that of the He Dsup-nucleosome complexes is consistent with the larger size of Rv Dsup (445 amino acid residues) compared to He Dsup (328 amino acid residues). Therefore, in the analysis of Rv Dsup and He Dsup, the relationship between the proteins (*Hashimoto and Kunieda, 2017*), the similarity of the gene arrangements (*Figure 5A*), and the conserved biochemical function (*Figures 2*, *5C and D*) lead to the conclusion that the two proteins are orthologs.

## Dsup protects chromatin from hydroxyl radical-mediated DNA cleavage in vitro

An intriguing property of Dsup is its ability to protect DNA in human cells from degradation that is induced either by X-ray irradiation or by treatment with hydrogen peroxide (*Hashimoto et al., 2016*). Because ionizing radiation and hydrogen peroxide both generate hydroxyl radicals as a major reactive oxygen species in cells (*Halliwell and Gutteridge, 1992*; *Stadtman, 1993*; *Riley, 1994*), we investigated whether purified Dsup affects hydroxyl radical-mediated DNA cleavage in a purified and defined biochemical system. To this end, we carried out hydroxyl radical-mediated DNA cleavage reactions in the absence or presence of Dsup.

In these experiments, we generated hydroxyl radicals by the Udenfriend modification of the Fenton reaction (*Udenfriend et al., 1954*; *Tullius et al., 1987*). Each of the hydroxyl radical reactions was carried out under the same conditions. In the absence of Dsup, the 3.3 kb plasmid DNA was mostly degraded to DNA fragments ranging from 100 to 1000 nt (*Figure 6A*). Notably, the extent of cleavage of free plasmid DNA was nearly the same as the amount of cleavage of chromatin. Thus, the presence of approximately one nucleosome per 200 bp DNA only slightly protects the DNA from hydroxyl radicals. These results are consistent with the previous observation that hydroxyl radicals can cleave DNA in nucleosomes (*Hayes et al., 1990*).

Upon addition of Rv Dsup, we observed substantial protection of free DNA as well as chromatin from hydroxyl radicals (*Figure 6A*). The Dsup-mediated protection was stronger with chromatin than

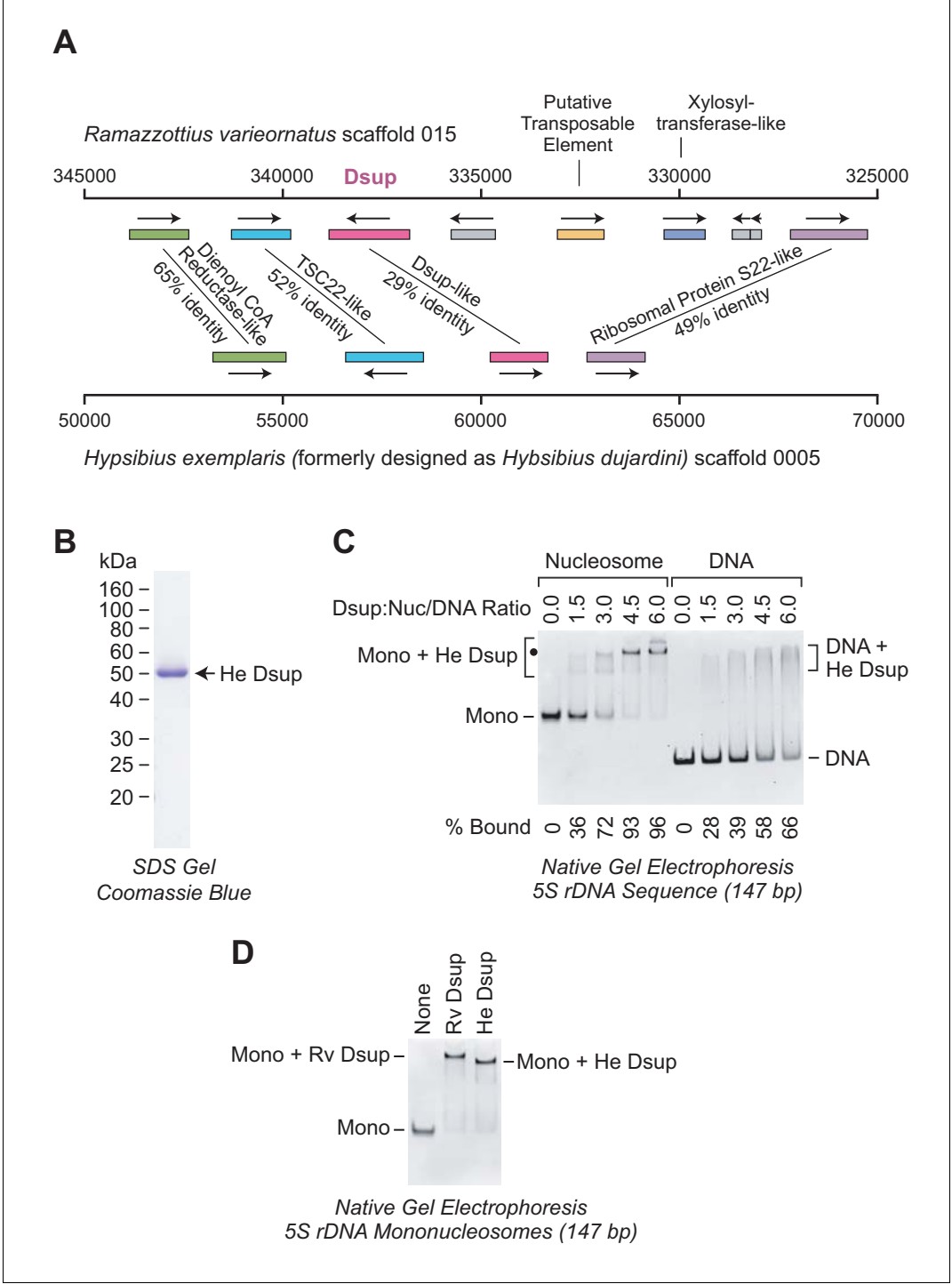

**Figure 5.** The Dsup-like protein from the tardigrade *Hypsibius exemplaris* is a nucleosome-binding protein. (**A**) Comparison of the genomic regions in the vicinity of the genes encoding *R. varieornatus* Dsup (Rv Dsup) and *H. exemplaris* Dsup-like protein (He Dsup). This diagram is based on sequences from *R. varieornatus* strain YOKOZUNA-1 scaffold 015 (*Hashimoto et al., 2016*) and *H. exemplaris* strain Z151 scaffold 0005 (*Yoshida et al., 2017*). The numbers correspond to the nucleotide positions in the scaffolds. The genomic regions in the vicinity of the *R. varieornatus* Dsup gene (accession number BAV59442.1) and *H. exemplaris* Dsup-like gene (accession number OQV24709.1) are shown. The protein sequences were analyzed by pairwise alignment with BLAST (*Altschul et al., 1997*). The % amino acid identities between the predicted proteins encoded by the corresponding genes are denoted. For each pair of homologous proteins, the number of identical amino acid

*Figure 5 continued*

residues divided by the length (in amino acid residues) of the region of homology is as follows: dienol CoA reductase-like (187/288); TSC22-like (155/300); Dsup (124/434); ribosomal protein S22-like (168/341). The difference in the % amino acid identity between the Rv Dsup and He Dsup proteins in *Hashimoto and Kunieda (2017)* (26.4%) versus this study (29%) is due to the use of different methods for the pairwise alignments. (B) Purification of recombinant He Dsup protein synthesized in *E. coli*. The recombinant He Dsup has an N-terminal His6-tag and a C-terminal FLAG tag. The purified protein was analyzed by 12% polyacrylamide-SDS gel electrophoresis and staining with Coomassie Brilliant Blue R-250. The sizes of molecular mass markers (in kDa) are indicated. (C) He Dsup binds preferentially to mononucleosomes relative to free DNA. Gel mobility shift experiments were performed as in *Figure 2A*. The positions of mononucleosomes (*Mono*), He Dsup-nucleosome complexes (*Mono + He Dsup*; the major shifted distinct band is denoted by a *black dot*), free DNA fragments (*DNA*), and Dsup-DNA complexes (*DNA + He Dsup*) are indicated. For each sample, the percent of the mononucleosomes or free DNA that was bound by Dsup was determined by quantitation of the unbound mononucleosomes or free DNA. (D) Comparison of the binding of Rv Dsup and He Dsup to mononucleosomes. Gel mobility shift experiments were performed with 147 bp 5S rDNA mononucleosomes, as in *Figures 2A* and *5C*, with 4.5 molecules of either Rv Dsup or He Dsup per nucleosome.

DOI: https://doi.org/10.7554/eLife.47682.008

with free DNA, possibly because the specific binding of Dsup to nucleosomes creates a highly resistant structure. In the presence of 4 molecules of Rv Dsup per 200 bp DNA (approximately 4 Dsup molecules per nucleosome) in chromatin, there was a considerable amount of full-length linear DNA remaining after hydroxyl radical-mediated cleavage. These results indicate that Rv Dsup is able to protect chromatin from the damaging effects of hydroxyl radicals. Because hydroxyl radicals react primarily with hydrogen atoms that are exposed in the minor groove of DNA (*Balasubramanian et al., 1998*), it appears likely that Dsup blocks access to the minor groove.

We additionally tested whether He Dsup and human TFIIB (a nuclear protein that does not bind to nucleosomes) can protect chromatin from hydroxyl radical-mediated DNA cleavage (*Figure 6B*). These experiments revealed that He Dsup, like Rv Dsup, is able to protect nucleosomal DNA from hydroxyl radical-mediated cleavage, and further support the conclusion that He Dsup is an ortholog of Rv Dsup. As seen in *Figure 6B*, we consistently observed that Rv Dsup provides slightly more protection against hydroxyl radicals than He Dsup. In contrast to Dsup, TFIIB did not confer any detectable protection of chromatin from hydroxyl radical-mediated DNA cleavage. Thus, these findings collectively indicate that Dsup binds to nucleosomes and protects DNA from cleavage by hydroxyl radicals (*Figure 6C*).

## A region of Dsup proteins with sequence similarity to vertebrate high mobility group N (HMGN) proteins is important for the binding of Dsup to nucleosomes

The homology between Rv Dsup and He Dsup led us to examine whether any of their conserved regions are related to protein sequences in organisms other than tardigrades. This analysis led to the identification of sequence similarity between a segment in the C-terminal region of the Dsup proteins and a conserved sequence in the nucleosome-binding domain of HMGN proteins (*Figure 7A*).

The HMGN proteins are abundant nucleosome-binding proteins that have been found only in vertebrates (for reviews, see: *Kugler et al., 2012*; *González-Romero et al., 2015*). They bind to two high affinity sites on nucleosomes in a manner that is independent of the DNA sequence. The nucleosome-binding domain of the HMGN proteins has been identified, and it contains a conserved core sequence, RRSARLSA, which is critical for the binding of HMGN proteins to nucleosomes (*Ueda et al., 2008*). Strikingly, both Dsup proteins contain a sequence that is similar to the RRSARLSA consensus (*Figure 7A*).

To examine whether the HMGN-like sequence in the Dsup proteins is important for their binding to nucleosomes, we generated two mutant versions of Rv Dsup (*Figure 7A*). The M1 mutant Dsup is a C-terminal deletion that removes the HMGN-like region, and the M2 mutant Dsup contains three R to E substitution mutations in the HMGN-like region. We purified the mutant Dsup proteins (*Figure 7B*) and then tested their ability to bind to mononucleosomes by gel mobility shift analysis

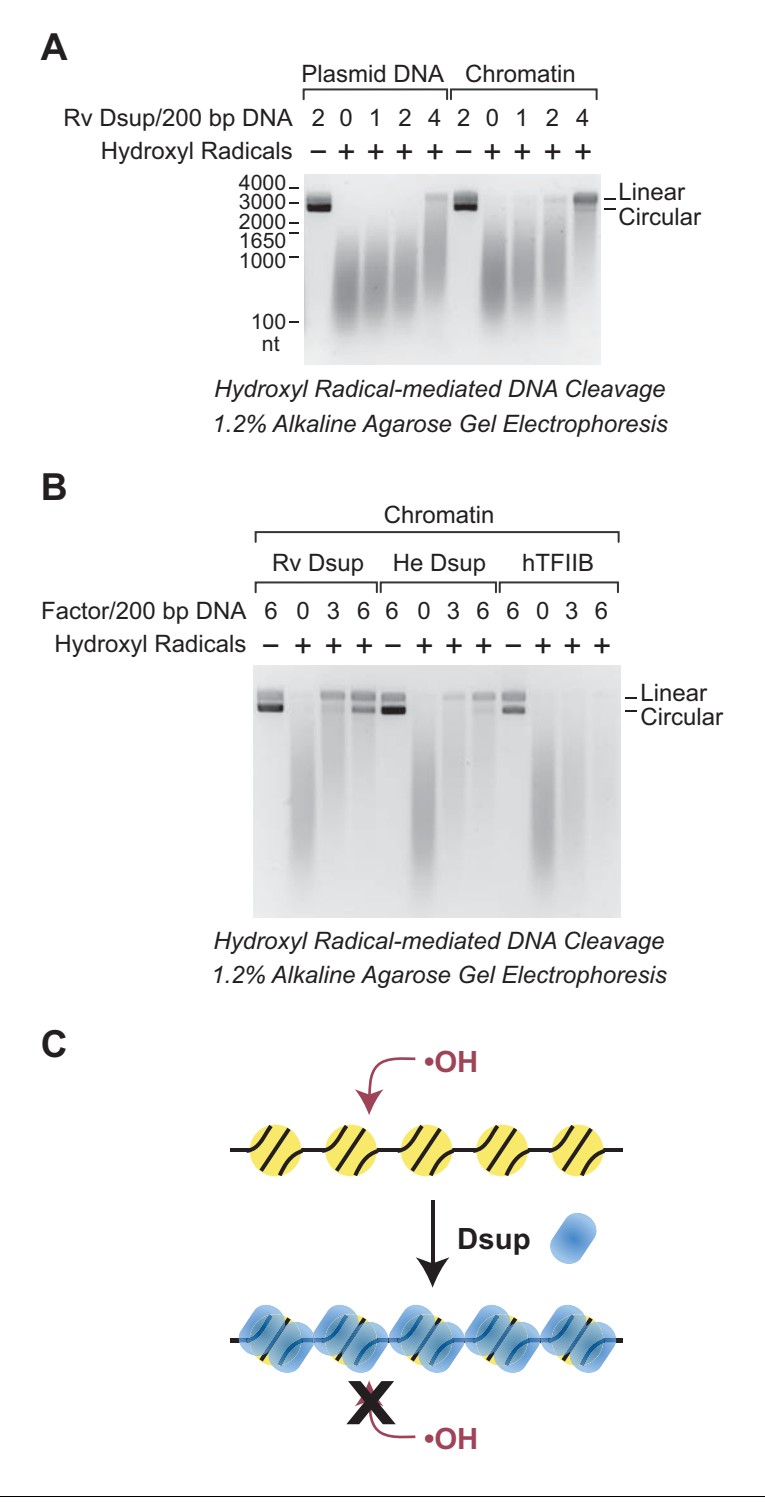

**Figure 6.** Dsup protects chromatin from hydroxyl radical-mediated DNA cleavage. Hydroxyl radical-mediated cleavage of DNA was carried out with plasmid pGIE-0 (3.3 kb) as either free DNA or chromatin in the absence or presence of the indicated factors. Where noted, hydroxyl radicals were omitted to test for the possible presence of nuclease activity in the factors. The reaction products were deproteinized, subjected to denaturing alkaline agarose gel electrophoresis, and visualized by staining with GelRed (Biotium). (**A**) Rv Dsup protects free DNA and chromatin from hydroxyl radical-mediated DNA cleavage. Hydroxyl radical DNA cleavage reactions were each performed under identical conditions with the indicated amounts of Rv Dsup. The full-length linear and circular DNA species are denoted. In addition, the positions of the size markers (in nt) are shown. (**B**) Hydroxyl radical-

*Figure 6 continued*
mediated DNA cleavage reactions of chromatin in the presence of Rv Dsup, He Dsup, and human TFIIB. Reactions were performed with chromatin as in A. Human TFIIB, a nuclear protein that does not bind to nucleosomes, was included as a control. For reference, the calculated molecular masses of the wild-type (untagged) proteins are as follows: Rv Dsup, 42.8 kDa; He Dsup, 33.0 kDa; human TFIIB, 34.8 kDa. (**C**) Model for Dsup binding to nucleosomes and protecting chromatin from hydroxyl radicals. In this model, Dsup binds specifically to nucleosomes and protects the DNA from damaging agents such as hydroxyl radicals via coverage of the chromatin with its disordered regions that are enriched in SAGK residues.
DOI: https://doi.org/10.7554/eLife.47682.009

(*Figure 7C*). These experiments revealed that the M1 mutant Dsup is nearly completely defective for binding to nucleosomes. Hence, the C-terminal region of Rv Dsup (from amino acid residues 360 to 445) is essential for nucleosome binding. In addition, the M2 mutant Dsup exhibits reduced binding to nucleosomes. Thus, the HMGN-like sequence in Dsup is important for its binding to nucleosomes.

We further investigated the ability of the mutant Dsup proteins to protect chromatin from hydroxyl radicals. These experiments revealed that the nucleosome-binding-deficient M1 mutant Dsup is nearly completely defective for protection of chromatin against hydroxyl radicals. We also observed that the M2 mutant Dsup is partially defective for protection against hydroxyl radicals. These data support the model that the binding of Dsup to nucleosomes is required to protect chromatin from hydroxyl radical-mediated DNA damage.

## Discussion

Here we have found that Dsup, a tardigrade-specific factor, is a nucleosome-binding protein that protects chromosomal DNA from hydroxyl radical-mediated cleavage. These findings provide a molecular explanation for the ability of Dsup to protect DNA in human cells from degradation by ionizing radiation or by treatment with hydrogen peroxide (*Hashimoto et al., 2016*). More generally, these results suggest that Dsup protects the tardigrade genome from hydroxyl radical-mediated damage under diverse conditions. For instance, protection of the tardigrade genome from hydroxyl radicals may contribute to survival after extended periods in the anhydrobiotic state (*Wełnicz et al., 2011*).

Notably, Dsup and histone H1 can bind simultaneously to nucleosomes (*Figure 4A*). Both *R. varieornatus* and *H. exemplaris* appear to contain histone H1, and it is thus likely that Dsup normally acts in conjunction with H1. In this regard, it is also notable that Dsup functions in histone H1-containing human cells (*Hashimoto et al., 2016*).

Rv Dsup and He Dsup are orthologs. In addition to their sequence similarity and related gene arrangements (*Figure 5A*), both Rv Dsup and He Dsup are nucleosome-binding proteins that protect chromatin from hydroxyl radicals (*Figures 5* and *6*). It had been previously suggested that Dsup is not present in *H. exemplaris* (*Yoshida et al., 2017*). If true, that would have implied that Dsup is not generally important for tardigrades. Instead, the identification of Dsup as a chromatin-protective protein in two species suggests that it is a key factor that contributes to the unique properties of tardigrades.

Unexpectedly, the Dsup proteins contain a region that exhibits sequence similarity to the conserved core of the nucleosome-binding domain of HMGN proteins (*Figure 7A*). Moreover, the HMGN-like sequence in Rv Dsup is functionally important for its ability to bind to nucleosomes (*Figure 7C*) and to protect chromatin from hydroxyl radicals (*Figure 7D*). Because the HMGN proteins have been found only in vertebrates (*González-Romero et al., 2015*), the origin of the HMGN-like sequences in the tardigrade Dsup proteins is, at present, a mystery. It is possible but not likely (see, for example, *Doolittle, 1994*) that the Dsup and HMGN proteins provide a very rare example of sequence convergence. In the future, detailed functional and evolutionary analyses might reveal the relation between these proteins.

The Dsup proteins are highly charged and largely disordered (*Hashimoto et al., 2016*; *Hashimoto and Kunieda, 2017*) (*Figure 1A*). With respect to the latter, examination of the amino acid composition of the Dsup proteins reveals that they are enriched in serine, alanine, glycine, and lysine (SAGK) residues. Rv Dsup contains more than 60% SAGK residues, and He Dsup has over 50%

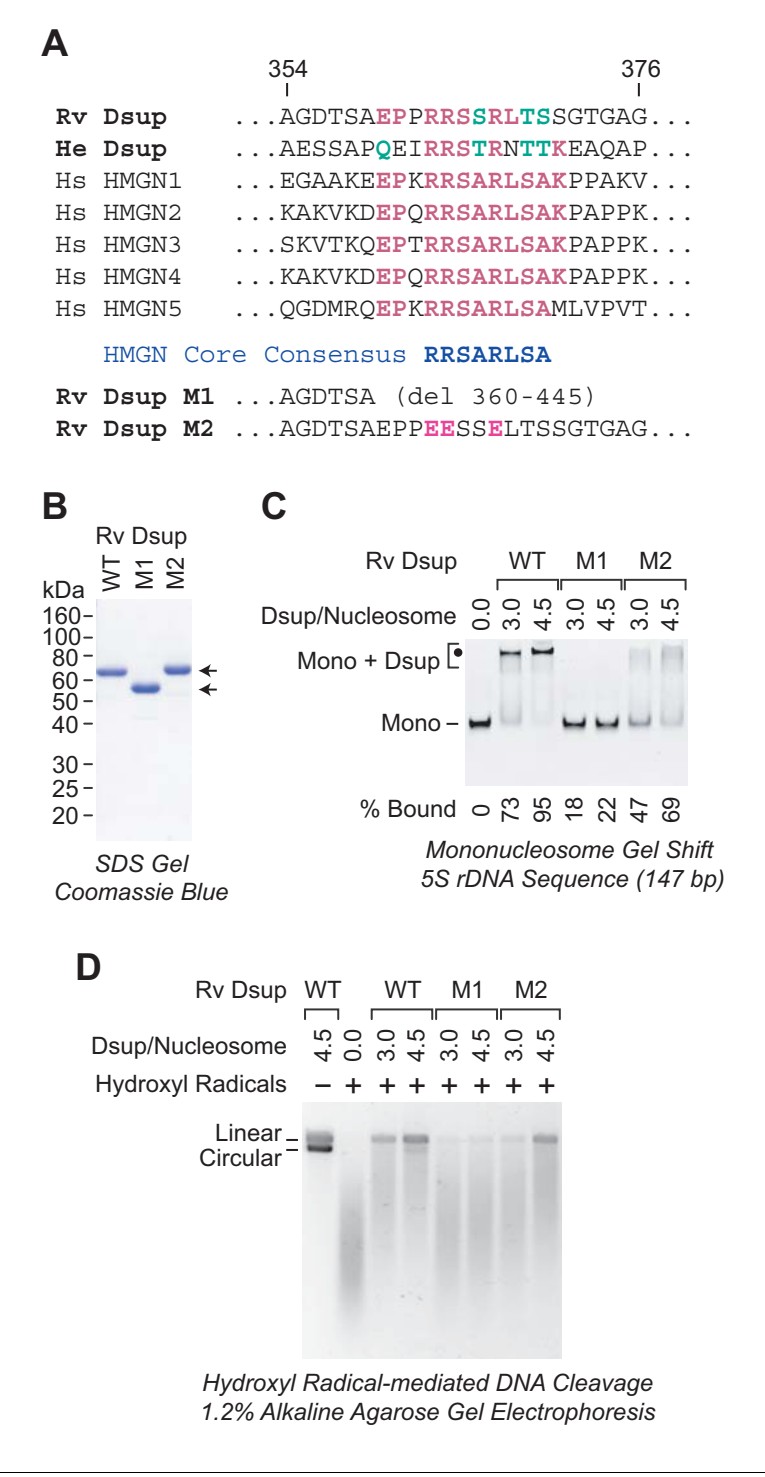

**Figure 7.** A region of sequence similarity between Dsup proteins and the nucleosome-binding domain of vertebrate HMGN proteins is important for the binding of Dsup to nucleosomes. (**A**) Alignment of Dsup proteins with the five human HMGN proteins as representative members of the HMGN protein family. Partial sequences of Rv Dsup, He Dsup, and the five human HMGN proteins are shown. The conserved HMGN consensus in the nucleosome-binding domain (*Ueda et al., 2008*) is indicated. Identical amino acid residues are highlighted in red type, and conserved amino acid substitutions (as in, for example, *Wu and Brutlag, 1996*) are in green type. The numbers indicate the amino acid residues in Rv Dsup. The amino acid sequences of Rv Dsup mutant M1 and mutant M2 are shown at the bottom. The modified residues in M2 are highlighted in pink type. (**B**) Purification of

*Figure 7 continued on next page*

*Figure 7 continued*

Rv Dsup mutant proteins M1 and M2. The purified proteins were analyzed by 12% polyacrylamide-SDS gel electrophoresis and staining with Coomassie Brilliant Blue R-250. The sizes of molecular mass markers (in kDa) are indicated. (C) The M1 and M2 mutant Dsup proteins bind less efficiently to mononucleosomes than wild-type Dsup. Gel mobility shift experiments were performed as in *Figure 2A*. The positions of mononucleosomes (*Mono*) and Dsup-nucleosome complexes (*Mono + Dsup*; the major shifted distinct band is denoted by a *black dot*) are indicated. For each sample, the percent of the mononucleosomes or free DNA that was bound by Dsup was determined by quantitation of the unbound mononucleosomes or free DNA. (D) The M1 and M2 mutant Dsup proteins are less effective at protecting chromatin from hydroxyl radical-mediated DNA cleavage than wild-type Dsup. Hydroxyl radical-mediated DNA cleavage reactions were performed with chromatin as in *Figure 6*. The reaction products were deproteinized, subjected to denaturing alkaline agarose gel electrophoresis, and visualized by staining with GelRed (Biotium). The full-length linear and circular DNA species are denoted.
DOI: https://doi.org/10.7554/eLife.47682.010

SAGK residues. SAGK are disorder-promoting amino acids (*Dunker et al., 2001*), and the SAGK residues may form a diffuse mass of protein that protects the chromosomal DNA from hydroxyl radical-mediated cleavage in a variety of conditions (*Figure 6C*). This model is consistent with the observation that tardigrades can survive high doses of ionizing radiation in the active as well as anhydriobiotic states (see, for example, *Jönsson et al., 2005*; *Horikawa et al., 2006*).

In conclusion, these studies reveal molecular aspects of Dsup function and suggest how this intriguing protein may contribute to the unique biological properties of tardigrades. A direct mechanism can be envisaged in which Dsup binds specifically to nucleosomes and protects the DNA from damaging agents such as hydroxyl radicals via coverage of the chromatin with its SAGK-rich disordered regions. In the future, the analysis of Dsup will reveal new insights into the physical and molecular bases of its functions and might further lead to its practical applications. For instance, it is possible that the DNA-protective properties of Dsup could be used to extend the longevity of cells. Thus, this unique protein from an extreme organism could be a new and powerful reagent in biological research.

## Materials and methods

Over the course of our studies, we developed three different methods for the purification of Dsup. We did not detect any difference in the activity of Dsup with any of the different purification methods described in this work. This observation is consistent with mostly unstructured nature of the Dsup proteins.

### Denaturing method for the purification of recombinant *R. varieornatus* Dsup

The cDNA sequence for Rv Dsup was synthesized with an N-terminal His6 tag and a C-terminal FLAG tag (Integrated DNA Technologies; San Diego, CA). The resulting DNA fragment was subcloned into pET21b to give pET21b-His6-Dsup-FLAG. For expression, freshly transformed *Escherichia coli* strain BL21(DE3) was grown in LB medium (2 L volume; 40 µg/mL ampicillin) at 37°C to $A_{600\ nm}$ of approximately 0.6 to 0.8. The synthesis of Dsup was induced by the addition of IPTG to 0.4 mM final concentration, and the culture was incubated at 18°C for 16 to 18 hr. The bacteria were collected by centrifugation (Fiberlite F9−4×1000y rotor; 6,000 rpm; 10 min; 18°C). Until stated otherwise, the next series of operations were carried out at room temperature (22°C). The pellet was resuspended in 20 mL of Denaturing Binding Buffer [20 mM sodium phosphate, pH 7.8, containing 8 M urea, 0.5 M NaCl, 5 mM 2-mercaptoethanol, 0.4 mM PMSF]. The cells were lysed with 3 × 15 strokes in a Wheaton Dounce homogenizer (B pestle), with a 10 min interval between each set of 15 strokes. The mixture was incubated for 1 hr and then subjected to centrifugation (Fiberlite F21S-8×50y rotor; 15,000 rpm; 20 min). The supernatant was collected and stored, and the pellet was resuspended with 15 mL of Denaturing Binding Buffer. The suspended pellet was dispersed with 15 strokes in a Wheaton Dounce homogenizer (B pestle), incubated for 1 hr, and subjected to centrifugation (Fiberlite F21S-8×50y rotor; 15,000 rpm; 20 min). The second supernatant was combined with the first supernatant, and the resulting mixture was incubated with 3 mL of Ni-NTA agarose

beads (Qiagen; Germantown, MD) for 2 hr on a rotating wheel at 4°C. At room temperature (22°C), the mixture was transferred into a Bio-Rad Econo-Pac chromatography column, and the resin was washed with 2 × 10 mL of Denaturing Wash Buffer [20 mM sodium phosphate, pH 6.0, containing 8 M urea, 0.5 M NaCl, 5 mM 2-mercaptoethanol, 0.4 mM PMSF]. The protein was eluted with Denaturing Elution Buffer [20 mM sodium phosphate, pH 4.0, containing 8 M urea, 0.5 M NaCl, 5 mM 2-mercaptoethanol, 0.4 mM PMSF] in 5 × 1 mL steps, with a 2 min interval between the addition of each mL of elution buffer. Unless stated otherwise, all subsequent operations were performed at 4°C. The eluate fractions were combined and dialyzed overnight in 4 L of PBS, and then incubated on a rotator wheel with 0.5 mL of anti-FLAG M2 Affinity Gel (Sigma-Aldrich; St. Louis, MO) for 3 to 4 hr. The beads were washed with 3 × 0.5 mL of Buffer A [20 mM Tris-HCl, pH 7.9, containing 150 mM NaCl, 2 mM MgCl₂, 0.2 mM EDTA, 15% (v/v) glycerol, 1 mM DTT], and protein was eluted with 4 × 0.1 mL portions of Buffer B [Buffer A containing 0.4 mg/mL FLAG peptide (Sigma-Aldrich; St. Louis, MO) and 0.5 mg/mL recombinant human insulin]. The protein fractions were frozen in liquid nitrogen and stored at −80°C. The Rv Dsup that was used in *Figures 1–4* was purified by this method.

## Nondenaturing method (and alternate denaturing method) for the purification of recombinant Dsup proteins from *H. exemplaris* and *R. varieornatus*

The nondenaturing method works very well for the purification of wild-type and mutant Dsup proteins from *R. varieornatus* and *H. exemplaris*. Here, we describe the synthesis and purification of He Dsup. The *E. coli* codon-optimized cDNA sequence for He Dsup was synthesized with a C-terminal FLAG tag (Integrated DNA Technologies; San Diego, CA) and inserted as an NcoI-XhoI fragment into pET21b-His6-Nco (*Kassavetis et al., 1998*). The resulting plasmid was transformed into Rosetta pLysS (MilliporeSigma). The cells were grown at 37°C in LB medium (0.5 L; 80 and 34 μg/mL, ampicillin and chloramphenicol, respectively) to an $A_{600 \text{ nm}}$ of 0.8, induced with IPTG to 0.5 mM final concentration, grown for an additional 2 hr, and harvested by centrifugation (Fiberlite F14S-6×250y rotor; 10,000 rpm; 15 min; 4°C). Unless stated otherwise, the remaining operations were carried out at 4°C. The cells (1.8 g) were resuspended in 12.6 mL Lysis Buffer [40 mM potassium phosphate, pH 7.6, 0.01% (v/v) NP-40 (Pierce), 0.1 mM EDTA, 580 mM NaCl, 10% (v/v) glycerol, 10 mM 2-mercaptoethanol, 1 μg/mL leupeptin, 1 μg/mL pepstatin, 0.5 mM PMSF, and 300 μg/mL lysozyme], incubated for 30 min on ice, sonicated (Branson Sonifier 450 with a 0.25 inch microtip; 20% output; 10 cycles of sonication of 13 s each; kept below 6°C by chilling in an ice-ethanol bath), and clarified by centrifugation (Fiberlite F21S-8×50y rotor; 18,000 rpm; 30 min). Dsup was purified from the soluble lysate supernatant fraction on a 0.5 mL Ni-NTA agarose column that was washed with 8 mL Buffer E [40 mM potassium phosphate, pH 7.6, 20 mM imidazole, 500 mM NaCl, 10% (v/v) glycerol, 10 mM 2-mercaptoethanol, 0.5 mM PMSF], pre-eluted with 300 μL Buffer E containing 200 mM imidazole, and eluted with 700 μL of the same buffer (yielding 5.8 mg/mL He Dsup and 3.6 mg/mL Rv Dsup). The samples were frozen in liquid nitrogen and stored at −80°C. For further purification, a portion of each sample (approximately 0.1 mg Dsup) was loaded onto a 300 μL anti-FLAG M2 agarose column that was pre-equilibrated in Buffer A2 [20 mM Tris-HCl, pH 7.9, containing 150 mM NaCl, 2 mM MgCl₂, 0.2 mM EDTA, 15% (v/v) glycerol, 1 mM 2-mercaptoethanol]. The column was left undisturbed for 5 min and washed with 2 mL Buffer A2. The Dsup was eluted by the addition of 250 μL (to give the pre-elution fraction) followed by 300 μL (elution fraction) Buffer A2 containing 300 μg/mL 3XFLAG peptide (ApexBio Technology). The samples were frozen in liquid nitrogen and stored at −80°C. The preparations of He Dsup and Rv Dsup in *Figures 6* and *7* were prepared in this manner. The method of preparation described in this paragraph is the most efficient means of obtaining high yields of pure Dsup proteins.

Alternate denaturing method. We also purified Dsup from the pellet of the bacterial cell lysate (prepared as described in the preceding paragraph) as follows. The pellet was resuspended in Buffer PG [40 mM potassium phosphate, pH 7.6, 20 mM imidazole, 6 M guanidine HCl, 10% (v/v) glycerol, 10 mM 2-mercaptoethanol, 1 μg/mL leupeptin, 1 μg/mL pepstatin, 0.5 mM PMSF] at room temperature (22°C), sonicated (Branson Sonifier 450 with a 0.25 inch microtip; 20% output; 4 cycles of sonication of 10 s each; kept below 6°C by chilling in an ice bath), and clarified by centrifugation (Fiberlite F21S-8×50y rotor; 18,000 rpm; 15 min; 4°C). The supernatant was loaded onto a 1 mL Ni-NTA agarose column, which was then washed with 11 mL Buffer E containing 6 M urea. The Dsup was renatured on the column with sequential 1.2 mL washes of Buffer E containing 5.0, 4.5, 4.0, 3.5, 3.0, 2.5,

2.0, 1.5, 1.0 and 0 M urea. The column was washed with 0.8 mL Buffer E containing 200 mM imidazole to give the pre-elution fraction, and protein was eluted with 1.3 mL of the same buffer to give the elution fraction (containing approximately 1 mg/mL Dsup). A portion of the elution fraction (approximately 0.1 mg Dsup) was loaded onto a 300 µL anti-FLAG M2 agarose column that was pre-equilibrated in Buffer A2 [20 mM Tris-HCl, pH 7.9, containing 150 mM NaCl, 2 mM MgCl$_2$, 0.2 mM EDTA, 15% (v/v) glycerol, 1 mM 2-mercaptoethanol]. The column was left undisturbed for 5 min and washed with 2 mL buffer A2. The Dsup was eluted by the addition of 250 µL (to give the pre-elution fraction) followed by 300 µL (elution fraction; 100 µg yield) Buffer A2 containing 300 µg/mL 3XFLAG peptide (ApexBio Technology). The protein fractions were frozen in liquid nitrogen and stored at −80°C. The Dsup proteins that were used in the gel mobility shift assays in *Figures 5*, *6* and *7* were prepared by the method described in this section.

It might be noted that Dsup proteins are hydrophilic and highly-charged proteins that bind weakly to SDS and thus migrate slowly on SDS-polyacrylamide gels relative to 'average' proteins such as the molecular mass markers. To confirm the authenticity of the Dsup proteins, we subjected wild-type *R. varieornatus* Dsup and wild-type *H. exemplaris* Dsup to LC-ESI-TOF-MS (liquid chromatography-electrospray ionization time-of-flight mass spectrometry). For each of the two proteins, the observed mass was within one mass unit of the predicted (calculated) mass. Thus, the Dsup proteins appear to be authentic.

## Salt dialysis reconstitution of mononucleosomes

Mononucleosomes were reconstituted by using the salt dialysis method (*Stein, 1989*; *Fei et al., 2015*) with purified recombinant *Drosophila* core histones (*Levenstein and Kadonaga, 2002*; *Khuong et al., 2017*) and the following DNA fragments. The 147 bp *Xenopus borealis* 5S rDNA fragment corresponds to the nucleosome positioning sequence that was mapped in *Fei et al. (2018)*. The sequence of the longer 181 bp *X. borealis* 5S rDNA fragment (with the central 147 bp positioning sequence indicated by lower case type ) is as follows: CAGGCTGTCAAGGCCGG gcttgtttcctgcctggggggaaaagaccctggcatggggaggagctgggcccccccccagaaggcagcacaaggggggaggaaaagt-cagccttgtgctcgcctacggccataccaccctgaaagtgcccgatatcgtctgatctcggaAGCCAAGCAGGGTCGGG. The 147 bp synthetic 601 nucleosome positioning sequence is described in *Lowary and Widom (1998)*. For nucleosome reconstitution, the DNA fragments were amplified by PCR and purified by 1.25% agarose gel electrophoresis and gel extraction (QIAquick Gel Extraction Kit; Qiagen; Germantown, MD). After reconstitution, the quality of the nucleosomes was assessed by nondenaturing 6% polyacrylamide gel electrophoresis. If necessary, the histone:DNA ratios in the reconstitution reactions were adjusted to achieve the efficient conversion of the DNA into mononucleosomes. If multiple nucleosomal species (due to the presence of mixture of differently positioned nucleosomes) were observed, the samples were heated at 58°C for 10 min to facilitate the movement of the differently positioned nucleosomes to the most stable location. The resulting mononucleosomes were stored at 4°C prior to use in the gel mobility shift experiments.

## Gel mobility shift analyses with mononucleosomes

Mononucleosomes (approximately 0.3 µM stock concentration; 60 nM final concentration) were combined with the indicated amounts of purified Dsup protein in HEG buffer [25 mM Hepes (K+), pH 7.6, 0.1 mM EDTA, 10% (v/v) glycerol] containing 100 mM KCl, and 100 ng bovine serum albumin in a total volume of 15 µL. The samples were incubated at 30°C for 30 min, and then loaded onto a nondenaturing 4.5% polyacrylamide gel that had been pre-equilibrated at 4°C and pre-run at 3.5 V/cm for 30 min. The gels were run at 4°C for 30 min at 3.5 V/cm and then for 2 hr at 8 V/cm. The nucleosomes were visualized by staining of the DNA with ethidium bromide. In experiments that involved testing Dsup and histone H1, the reaction medium additionally included 0.05% (v/v) NP-40 (Pierce) and 2 mM MgCl$_2$. The histone H1 was purified from *Drosophila* embryos by the method of *Croston et al. (1991)*. To ensure reproducibility of the data, each of the reported experiments was performed a minimum of two independent times with different preparations (*i.e.*, biologically distinct samples) of Dsup.

## ACF-mediated assembly of periodic nucleosome arrays

The ATP-dependent assembly of periodic nucleosome arrays with the ACF motor protein and the dNLP core histone chaperone was carried out essentially as described in *Fyodorov and Kadonaga (2003)* and *Khuong et al. (2017)*. A standard reaction included purified *Drosophila* core histones (0.35 µg), purified *Drosophila* ACF (10 nM), purified *Drosophila* dNLP (1.8 µg), 2 mM ATP, purified *Drosophila* topoisomerase I (ND423) (1 nM), and pGIE-0 plasmid DNA (0.35 µg) (*Pazin et al., 1994*) in a total volume of 70 µL. Purified Dsup and histone H1 were included where indicated. The final KCl concentration was 100 mM for samples that were analyzed by the DNA supercoiling and partial MNase assays and 50 mM for samples that were subjected to extensive (limit) MNase digestion. The assembly reactions were incubated at 27°C for 90 min. The partial MNase, extensive MNase, and DNA supercoiling assays were performed as described by *Fyodorov and Kadonaga (2003)* and *Khuong et al. (2017)*. To ensure reproducibility of the data, each of the reported experiments was performed a minimum of two independent times with different preparations (*i.e.*, biologically distinct samples) of Dsup.

## Hydroxyl radical-mediated cleavage of nucleosomal DNA

Nucleosomes were reconstituted onto plasmid pGIE-0 (3269 bp) by the salt dialysis method (*Stein, 1989*; *Fei et al., 2015*) with 16.4 pmol *Drosophila* core histone octamers per pmol pGIE-0 (average of one nucleosome per 200 bp DNA). The final dialysis buffer was Buffer TEN [10 mM Tris-HCl, pH 8.0, 1 mM EDTA, 0.01% (v/v) NP-40 (Pierce)] containing 50 mM NaCl. Proteins were diluted into Buffer TEN containing 50 mM NaCl prior to use. For each series of reactions with a particular factor (such as Rv Dsup, He Dsup, or human TFIIB), the protein and its corresponding storage buffer were diluted in parallel and then used in a manner that ensured that each reaction was carried out under identical buffer conditions. We also note that the highest concentration of glycerol in any of the reactions was 0.014% (v/v). At this concentration of glycerol, we did not detect quenching of the hydroxyl radical cleavage reactions by the buffer medium. Hydroxyl radical-mediated cleavage of DNA was carried out essentially as described by *Tullius et al. (1987)*. Briefly, nucleosomal pGIE-0 (162 fmol) along with the indicated amount of added protein were incubated at room temperature (22°C) for 10 min in 90 µL Buffer TEN containing 50 mM NaCl. Then, in each tube, three separate droplets of 60 mM ascorbic acid (3.3 µL), 100 µM Fe(II) and 200 µM EDTA (3.3 µL), and 0.3% (v/v) $H_2O_2$ (3.3 µL) were deposited onto different locations on the tube wall (suspended above the chromatin sample at the bottom of the tube), mixed together, and vortexed into the 90 µL binding reaction. The reaction was stopped 2 min later by the addition of 10 µL of 200 mM thiourea. The DNA was extracted with phenol-CHCl₃-isoamyl alcohol, precipitated with ethanol, and resuspended in 10 µL of 50 mM NaOH-1 mM EDTA and 2 µL of Purple Loading Dye (New England Biolabs). Alkaline agarose gel electrophoresis was performed as described by *Shin and Day (1995)*, and the DNA was stained with GelRed (Biotium) in 50 mM Tris-HCl, pH 6.8. To ensure reproducibility of the data, each of the reported experiments was performed a minimum of two independent times with different preparations (*i.e.*, biologically distinct samples) of Dsup.

## Acknowledgements

This paper is dedicated to Professor Russell F Doolittle in recognition of his many contributions to science, particularly with regard to the structure and evolution of proteins. We are very grateful to Russell F Doolittle for carrying out the evolutionary analysis of Dsup from R varieornatus and H. exemplaris as well as for advice in the preparation of the manuscript. We thank E Peter Geiduschek, Long Vo ngoc, Cassidy Yunjing Huang, and Bailey Munro for critical reading of the manuscript. GC-B is a Pew Latin American Fellow in the Biomedical Sciences. JTK. is the Amylin Endowed Chair in Life-sciences Education and Research. This work was supported by NIH/NIGMS grant R35 GM118060 to JTK.

## Additional information

### Funding

| Funder | Grant reference number | Author |
|---|---|---|
| National Institutes of Health | R35 GM118060 | James T Kadonaga |

The funders had no role in study design, data collection and interpretation, or the decision to submit the work for publication.

### Author contributions

Carolina Chavez, Grisel Cruz-Becerra, George A Kassavetis, Conceptualization, Resources, Validation, Investigation, Visualization, Methodology, Writing—original draft, Writing—review and editing; Jia Fei, Conceptualization, Resources, Supervision, Validation, Visualization, Methodology, Writing—original draft, Project administration, Writing—review and editing; James T Kadonaga, Conceptualization, Supervision, Funding acquisition, Validation, Visualization, Methodology, Writing—original draft, Project administration, Writing—review and editing

### Author ORCIDs

Grisel Cruz-Becerra ⓘ http://orcid.org/0000-0001-6297-4132
George A Kassavetis ⓘ https://orcid.org/0000-0001-6857-5271
James T Kadonaga ⓘ https://orcid.org/0000-0002-2075-9458

### Decision letter and Author response

Decision letter https://doi.org/10.7554/eLife.47682.013
Author response https://doi.org/10.7554/eLife.47682.014

## Additional files

### Supplementary files

• Transparent reporting form
DOI: https://doi.org/10.7554/eLife.47682.011

### Data availability

All data generated in this study are included in the manuscript.

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
