## [Decision Letter]

Thank you for submitting your article "The tardigrade damage suppressor protein binds to nucleosomes and protects DNA from hydroxyl radicals" for consideration by *eLife*. Your article has been reviewed by four peer reviewers, one of whom is a member of our Board of Reviewing Editors, and the evaluation has been overseen by Jessica Tyler as the Senior Editor. The reviewers have opted to remain anonymous.

The reviewers have discussed the reviews with one another and the Reviewing Editor has drafted this decision to help you prepare a revised submission.

Summary:

This paper reports on the remarkable biochemical properties of tardigrade damage suppressor (Dsup) proteins which bind to single nucleosomes reconstituted on 5S and 601 sequences as well as nucleosome arrays, conferring protection from nuclease and hydroxyl radical cleavage. Authors show that Dsup does not interfere with ACF-mediated nucleosome assembly nor with binding of the linker histone H1. They also provide functional evidence for an orthologous protein from a different tardigrade species, highlighting the potentially general function of Dsup proteins in the enabling resistance of tardigrades to ionizing radiation. Overall, the quality of biochemical evidence presented is in keeping with the high standards of the Kadonaga laboratory. The study of these proteins has exciting implications for the prospects of engineering organisms to withstand extreme physical conditions and conditions that otherwise create genomic instability problems associated with aging and cancer.

Essential revisions:

1) The most critical issue for revision concerns the absence of a natural frame of reference to interpret the observed influence of Dsup on nucleosome binding and protection from hydroxyl radical cleavage. What is the Dsup-to-nucleosome stoichiometry in tardigrades? If Dsup protects tardigrade DNA from damage it would seem that the number of Dsup molecules would have to be on the same order as the number of nucleosomes. The authors should comment on the relative abundance of Dsup and nucleosomes in cells. This information is essential to interpret the findings that high Dsup to nucleosome ratios are needed for cleavage protection in the authors' experiments. The relative stoichiometry may be in the tardigrade literature, if so, authors should cite and discuss.

2) While the manuscript documents a highly interesting and potentially novel function for a distinctive nucleosome-binding protein, the current level of mechanistic understanding is limited. How do Dsups bind to nucleosomes in a way that blocks hydroxyl radical cleavage? Do Dsups cover major or minor DNA grooves or the phosphate backbone? What is the role of the SAGK motif in contacting nucleosomal DNA? Is DNA curvature sufficient for preferential binding to the NCP versus linker DNA, or are histones also involved in binding? The provision of additional exploration of mechanisms along some of these lines and inclusion of insights gained is necessary to justify accepting the current manuscript for *eLife*.

3) In describing Figure 2A the authors state, "These experiments revealed that Rv Dsup binds with a higher affinity to nucleosomes than to free DNA." Although this appears to be the case, the conclusion would be stronger if the authors quantitated the gels and estimated the affinities for Dsup binding nucleosomes and free DNA.

4) The DNA protection activity (in Figure 6B) is really intriguing data. Both RvDsup and HdDsup exhibited a substantial protection of DNA against hydroxyl radicals, which is really great. However, TFIIB is not a good control for this kind of experiment, because it is not a nucleosome-binding protein, and any nucleosome-binding factor may provide some protection of DNA against damage. Therefore, a different protein which is capable of binding nucleosomes should be examined as a comparison counterpart to Dsups. Also, please provide a clear description on the protein prep. for the hydroxy radical protection assay.

5) Previous work indicated that the C-terminus is sufficient for binding to DNA and entry to the nucleus, but is not sufficient for genome protection in vivo. Is the C-terminus sufficient for nucleosome-binding in vitro, or does nucleosome protection require additional domains than required than for binding DNA? Do Dsup proteins self-aggregate on nucleosome arrays?

6) *Hypsibius dujardini*, especially the strain used for the genome/molecular analyses, was recently reidentified as a new species, *Hypsibius exemplaris*. The authors should use the new species name with a short explanation on the re-identification of the species. cf.) Gąsiorek et al., 2018.

---

## [Author Response]

We thank the reviewers for the positive and enthusiastic assessment of this work. Upon receipt of the reviews, we suggested a strategy for addressing the issues that were raised in the Essential Revisions, and this plan was approved by editors at *eLife*.

Fortunately, in the course of our analysis of the structure and function of Dsup in the revision of this work, we made the remarkable discovery that a conserved region of Dsup exhibits sequence similarity to the core consensus nucleosome-binding domain of vertebrate HMGN proteins (new Figure 7A). Importantly, by mutational analysis, we found that the HMGN-like sequence in Dsup is functionally important for nucleosome binding (new Figure 7C) and protection of chromatin from hydroxyl radicals (new Figure 7D). These results provide the following: (i) strong evidence that the binding of Dsup is required for its protection of DNA against hydroxyl radicals; (ii) identification of a key region of Dsup that is important for binding to nucleosomes; and (iii) a surprising and intriguing relation between the tardigrade Dsup protein and HMGN proteins, which have been found only in vertebrates. These findings add an exciting new dimension to this work.

Importantly, these finding address the main conceptual issues raised by the reviewers. We also addressed the technical matters, such as the quantitation of the gel mobility shift data (new Figure 2—figure supplement 1), the re-naming of *Hypsibius dujardini* as *Hypsibius exemplaris*, and the revision of the Materials and methods.

Essential revisions:1) The most critical issue for revision concerns the absence of a natural frame of reference to interpret the observed influence of Dsup on nucleosome binding and protection from hydroxyl radical cleavage. What is the Dsup-to-nucleosome stoichiometry in tardigrades? If Dsup protects tardigrade DNA from damage it would seem that the number of Dsup molecules would have to be on the same order as the number of nucleosomes. The authors should comment on the relative abundance of Dsup and nucleosomes in cells. This information is essential to interpret the findings that high Dsup to nucleosome ratios are needed for cleavage protection in the authors' experiments. The relative stoichiometry may be in the tardigrade literature, if so, authors should cite and discuss.

With regard to this question, Supplementary Figure 10 (see also Supplementary Tables 4 and 5) of Hashimoto et al., 2016, shows an SDS gel of the chromatin (Ch) and soluble nuclear (NS) fractions from *R. varieornatus*. A Dsup peptide is in band B1, which is a major protein band in the chromatin fraction that is not seen in the soluble nuclear fraction. Dsup can also be seen to colocalize with chromatin in *R. varieornatus* embryos by immunofluorescence (Supplementary Figure 12). These methods do not, however, provide a quantitative assessment of the number of Dsup molecules per nucleosome in tardigrades. Such data are not yet available, and, given the challenges in analyzing pure tardigrade tissue, it may be many years before such data are obtained. Nevertheless, Dsup is sufficiently abundant to be identified in a chromatin fraction and to be clearly visualized by immunofluorescence. It would certainly be a point of concern if it could not be detected in tardigrades. Because it would be technically extremely challenging to obtain high quality quantitative data in tardigrades, we felt that it would not be reasonably possible to obtain such data for this paper or to make a meaningful comment on the existing data. Fortunately, the *eLife* editors agreed with this assessment.

2) While the manuscript documents a highly interesting and potentially novel function for a distinctive nucleosome-binding protein, the current level of mechanistic understanding is limited. How do Dsups bind to nucleosomes in a way that blocks hydroxyl radical cleavage? Do Dsups cover major or minor DNA grooves or the phosphate backbone? What is the role of the SAGK motif in contacting nucleosomal DNA? Is DNA curvature sufficient for preferential binding to the NCP versus linker DNA, or are histones also involved in binding? The provision of additional exploration of mechanisms along some of these lines and inclusion of insights gained is necessary to justify accepting the current manuscript for eLife.

Four points are as follows.

i) Hydroxyl radicals react with the hydrogen atoms on the deoxyribose backbone with the following preference: 5' H > 4' H > 3' H ~ 2' H ~ 1' H [Balasubramanian et al., 1998]. Because the hydrogen atoms on the 5' and 4' carbons are exposed in the minor groove of DNA, it appears that Dsup blocks access to the DNA minor groove. This point is now included in the paper as follows:

"Because hydroxyl radicals react primarily with hydrogen atoms that are exposed in the minor groove of DNA (Balasubramanian et al., 1998), it appears likely that Dsup blocks access to the minor groove."

ii) There is no "SAGK motif". Dsup is highly enriched in S, A, G, and K residues (>60% of all aa residues in *R. varieornatus* Dsup), but they do not exist as an SAGK motif. To minimize the possibility that readers might inadvertently think that there is an SAGK motif, we changed (in the Discussion) "… the SAGK-rich regions.…" to "… the SAGK residues.…"

iii) The first two items address the specific questions raised in point 2 except for the issue of DNA curvature. From a broader perspective, it might be noted that additional data on the possible role of DNA curvature would not change the main conclusions of the paper. I was one of the original BREs for *eLife*, and I served as a BRE for three years (and then briefly as a Senior Editor). In our training sessions, we were told that *eLife* is not interested in requesting additional experiments that do not change the main conclusions of the paper. As far as I understand, this policy has not changed.

iv) Lastly, we did address the mechanism of Dsup binding to nucleosomes. These experiments are described in greater detail below in the response to Essential Points 4 and 5. In brief, we identified a conserved region of Dsup that is related to the nucleosome-binding region of vertebrate HMGN proteins. Mutation of this region of Dsup resulted in a reduction or near complete loss of the nucleosome binding and hydroxyl radical protection activities. It is interesting and intriguing that Dsup may bind to nucleosomes by a mechanism that is related to the binding of vertebrate HMGN proteins to nucleosomes.

3) In describing Figure 2A the authors state, "These experiments revealed that Rv Dsup binds with a higher affinity to nucleosomes than to free DNA." Although this appears to be the case, the conclusion would be stronger if the authors quantitated the gels and estimated the affinities for Dsup binding nucleosomes and free DNA.

We have now provided quantitative data. For the primary gel shift experiments (as in Figures 2A and 2B), we quantitated three independent experiments and provided graphs (mean +/– standard deviation of the percent bound DNA/nucleosomes versus Dsup:DNA/nucleosome molar ratio) in the new Figure 2—figure supplement 1. These results further confirm the conclusions. We additionally provided quantitation of the gel shift data in Figures 5C and 7C. We did not attempt to estimate the absolute affinities, as the main point was to address whether Dsup binds with higher affinity to nucleosomes than to free DNA (and the absolute affinities would not alter the conclusions of the paper).

4) The DNA protection activity (in Figure 6B) is really intriguing data. Both RvDsup and HdDsup exhibited a substantial protection of DNA against hydroxyl radicals, which is really great. However, TFIIB is not a good control for this kind of experiment, because it is not a nucleosome-binding protein, and any nucleosome-binding factor may provide some protection of DNA against damage. Therefore, a different protein which is capable of binding nucleosomes should be examined as a comparison counterpart to Dsups. Also, please provide a clear description on the protein prep. for the hydroxy radical protection assay.

With regard to the issue of hydroxyl radical protection and the use of TFIIB as a control, we felt that it would be best to address these comments by further testing whether the binding of Dsup to nucleosomes is required for protection against hydroxyl radicals. Based on the data in the submitted paper, it was formally possible that Dsup generally protects DNA from hydroxyl radicals by a mechanism that does not involve binding to nucleosomes. To address this important question, we generated a mutant version of Dsup (M1 mutation) that has greatly reduced binding to nucleosomes (see new Figures 7A, 7B, and 7C). Importantly, we then found that the M1 mutant Dsup is severely defective in its ability to protect chromatin from hydroxyl radicals (Figure 7D). Thus, in this manner, we were able to show that Dsup-mediated protection against hydroxyl radicals depends upon the binding of Dsup to chromatin.

With regard to the protein purification, we revised and clarified the Materials and methods so that the exact method of preparation of Dsup for the hydroxyl radical protection assay is clearly indicated.

*5) Previous work indicated that the C-terminus is sufficient for binding to DNA and entry to the nucleus, but is not sufficient for genome protection* in vivo*. Is the C-terminus sufficient for nucleosome-binding* in vitro*, or does nucleosome protection require additional domains than required than for binding DNA? Do Dsup proteins self-aggregate on nucleosome arrays?*

In Hashimoto et al., 2016 (Supplementary Figure 14), it was shown that the C-terminal region of Dsup (an N-terminal deletion lacking N-terminal residues 2-207 out of 445 aa residues) can interact with DNA and localize to the nucleus. However, in the genome protection assay, Hashimoto et al., 2016, tested only the N-terminal region (i.e., a C-terminal deletion) and not the C-terminal region (Supplementary Figure 17). Thus, it was not shown that the C-terminal region "is not sufficient for genome protection in vivo". Therefore, it is not possible to follow up on point 5 as stated because the genome protection data by the C-terminal region do not exist.

Importantly, we were nevertheless able to address questions that are related to the issues raised by the reviewers in this point. As shown in the new Figure 7, we made the surprising observation that a region that is conserved between Rv Dsup and He Dsup exhibits a striking sequence similarity to the central core region of the vertebrate HMGN proteins (new Figure 7A). This region is near the C-terminus of the Dsup proteins. We therefore generated a C-terminal deletion of Dsup (M1; deletion of amino acid residues 360-445) as well as a triple R to E amino acid substitution mutant Dsup (termed M2) (new Figure 7A). We then found that the M1 mutant is nearly completely defective for binding to nucleosomes and that the M2 mutant is moderately defective for nucleosome binding (new Figure 7C). We have thus identified a conserved region of Dsup that is important for binding to nucleosomes and protection against hydroxyl radicals (new Figure 7D). Intriguingly, this region is related to the core nucleosome-binding domain of vertebrate HMGN proteins. Because the HMGN nucleosome-binding domain has not been found outside of vertebrates, it is really quite unexpected and interesting to find a related and functionally important sequence in tardigrades.

With regard to the last point about aggregation of Dsup on chromatin, it should be noted that there is no evidence of aggregation of Dsup on nucleosome arrays, as seen in the partial MNase digestion analyses (Figure 3B and Figure 4C) as well as in the extensive MNase digestion analysis (Figure 3C) of nucleosome arrays containing Dsup.

6) Hypsibius dujardini, especially the strain used for the genome/molecular analyses, was recently reidentified as a new species, Hypsibius exemplaris. The authors should use the new species name with a short explanation on the re-identification of the species. cf.) Gąsiorek P, et al., 2018.

Thank you for the update. We changed "*H. dujardini*" to "*H. exemplaris*" throughout the text and figures and cited the relevant paper (Gąsiorek et al., 2018). In the Introduction, we included the following statement:

"Note: the strain of tardigrades that was designated as *Hypsibius dujardini* in Yoshida et al., 2017, has since been found to be a new species that is now termed *Hypsibius exemplaris* (Gąsiorek et al., 2018). We will use the new terminology in this paper."